# Long-Term Immunity against SARS-CoV-2 Wild-Type and Omicron XBB.1.5 in Indonesian Residents after Vaccination and Infection

**DOI:** 10.3390/antib13030072

**Published:** 2024-09-02

**Authors:** Ammar Abdurrahman Hasyim, Akihiko Sakamoto, Kyouhei Yamagata, Kartika Hardianti Zainal, Desi Dwirosalia Ningsih Suparman, Ika Yustisia, Marhaen Hardjo, Syahrijuita Kadir, Mitsuhiro Iyori, Shigeto Yoshida, Yenni Yusuf

**Affiliations:** 1Master Program of Biomedical Science, Graduate School of Hasanuddin University, Makassar 90245, Indonesia; karismananda23p@student.unhas.ac.id (K.); ikayustisia@pasca.unhas.ac.id (I.Y.); marhaenhardjo@gmail.com (M.H.); syahrijuita@med.unhas.ac.id (S.K.); 2Laboratory of Vaccinology and Applied Immunology, Kanazawa University, Kanazawa 920-1192, Japan; ammarhasyim26@gmail.com (A.A.H.); aron@stu.kanazawa-u.ac.jp (K.Y.); kartikahardianti@gmail.com (K.H.Z.); shigeto@p.kanazawa-u.ac.jp (S.Y.); 3Department of Biochemistry, Faculty of Medicine, Bosowa University, Makassar 90232, Indonesia; desi.ddrns@gmail.com; 4Department of Biochemistry, Faculty of Medicine, Hasanuddin University, Makassar 90245, Indonesia; 5Research Institute of Pharmaceutical Sciences, Musashino University, Nishitokyo 202-8585, Japan; m-iyori@musashino-u.ac.jp; 6Department of Parasitology, Faculty of Medicine, Hasanuddin University, Makassar 90245, Indonesia

**Keywords:** immune persistence, Omicron XBB.1.5, neutralizing capacity, antibodies, COVID-19 vaccines

## Abstract

In the post-pandemic era, evaluating long-term immunity against COVID-19 has become increasingly critical, particularly in light of continuous SARS-CoV-2 mutations. This study aimed to assess the long-term humoral immune response in sera collected in Makassar. We measured anti-RBD IgG levels and neutralization capacity (NC) against both the Wild-Type (WT) Wuhan-Hu and Omicron XBB.1.5 variants across groups of COVID-19-vaccinated individuals with no booster (NB), single booster (SB), and double booster (DB). The mean durations since the last vaccination were 25.11 months, 19.24 months, and 16.9 months for the NB, SB, and DB group, respectively. Additionally, we evaluated the effect of breakthrough infection (BTI) history, with a mean duration since the last confirmed infection of 21.72 months. Our findings indicate fair long-term WT antibody (Ab) titers, with the DB group showing a significantly higher level than the other groups. Similarly, the DB group demonstrated the highest anti-Omicron XBB.1.5 Ab titer, yet it was insignificantly different from the other groups. Although the level of anti-WT Ab titers was moderate, we observed near-complete (96–97%) long-term neutralization against the WT pseudo-virus for all groups. There was a slight decrease in NC against Omicron XBB.1.5 compared to the WT among all groups, as DB group, SB group, and NB group showed 80.71 ± 3.9%, 74.29 ± 6.7%, and 67.2 ± 6.3% neutralization activity, respectively. A breakdown analysis based on infection and vaccine status showed that booster doses increase the NC against XBB.1.5, particularly in individuals without BTI. Individuals with BTI demonstrate a better NC compared to their counterpart uninfected individuals with the same number of booster doses. Our findings suggest that long-term immunity against SARS-CoV-2 persists and is effective against the mutant variant. Booster doses enhance the NC, especially among uninfected individuals.

## 1. Introduction

The COVID-19 pandemic transitioned from a pandemic to an endemic status in May 2023 [1]. Despite this change, ongoing sequelae and threats pose significant challenges due to the reported mutations that can cause re-infection and subsequent consequences. The WHO report in December 2023 indicated a 53% increase in cases and a 51% increase in ICU admissions [2]. Furthermore, several studies have shown that post-COVID conditions, also known as long COVID-19, can cause severe and debilitating symptoms such as chronic fatigue, persistent respiratory issues, persistent muscle pain, sleep disorders, and cognitive impairments like brain fog [3]. Therefore, alertness against the potential for infection or re-infection with COVID-19 remains crucial in protecting public health. Given the uncertainty surrounding SARS-CoV-2, we must remain vigilant against the threat of this virus’ mutations.

The Omicron variant of SARS-CoV-2 has demonstrated increased transmissibility and the ability to evade immunity derived from vaccines, although the first-generation vaccines still protect against severe illness and death [4]. The effectiveness of vaccines against Omicron infection is generally lower compared to previous variants, with protection decreasing rapidly over time [4,5,6]. A new COVID-19 vaccine containing targeting Omicron strain antigens was approved in 2023 to face this problem [7,8,9].

Vaccination in Indonesia started in January 2021, using two doses of Coronavac^®^, the whole inactivated virus vaccine, at 4-week intervals. According to the Indonesian Ministry of Health, 86% of the population has received one dose, and 74% received the second dose. The first booster was introduced in August 2021 and reached 39% of residents in December 2022, while the second booster only covered 2% of the population in early 2023 [10]. The vaccination coverage leveled up since then, and there has been no significant increase in the number of individuals receiving vaccination since May 2022 [11]. Thus, the time gap between vaccination programs in Indonesia and the emergence of the Omicron strain can potentially reduce the level of immunity in the Indonesian population against the variant.

The continued evaluation of immune responses targeting circulating variants of SARS-CoV-2 is important to guide future vaccination strategies. Recently, at the end of 2023, the Indonesian population was struck by the XBB.1.5 variant of the Omicron strain, which caused the deaths of the elderly after a long pause in COVID-19-associated mortality [12]. This strain—emerging as the dominant strain in several countries—is more transmissible and has a superior capability to escape immunity [13,14]. Thus, several questions arose regarding the immune responses against this variant, such as the persistence of the humoral response induced by vaccinations and the effectivity of vaccine-induced antibodies in neutralizing the virus, particularly for the circulating Omicron XBB.1.5 strain. In addition, will we need to introduce a vaccine containing the Omicron antigen? This study was conducted to address those questions and to predict the vulnerability of the population to re-infection by emerging mutant strains of SARS-CoV-2.

## 2. Materials and Methods

### 2.1. Study Design and Study Subject

We conducted a cross-sectional study by recruiting individuals with different vaccination statuses by the number of booster doses using purposive sampling in Makassar, Indonesia’s capital city of South Sulawesi Province, from November 2023 to January 2024 (Table 1). The inclusion criteria were those above 17 years old who had received two doses of the inactivated whole SARS-CoV-2 virus vaccine CoronaVac^®^ from Sinovac at a 4-week interval as the primary vaccination. This study has been approved by the Ethics Committee of Hasanuddin University (Approval Number 182/UN4.6.4.5.31/PP36/2024).

Blood was withdrawn after the subjects signed the informed consent form. We had also recorded the vaccination and infection history since January 2022. The collected blood was centrifuged for serum separation. All sera were kept at −80 °C before being subjected to any experiments.

We recruited 113 subjects for the study from December 2023 to January 2024. The distribution of participants according to their vaccination status is summarized in Table 2. The proportion of participants who received two booster doses (DB group) was 41.59%, while those who received no booster dose (NB group) comprised 32.74% of the total subjects. The age varied across groups, with the highest median age observed in the participants with two booster doses (34.96 y.o, interval of 20–40 y.o). The infection status differed among the vaccination groups, with the highest percentage of confirmed infection history in the participants with two booster doses (29.78%). We recorded the confirmed breakthrough infection (BTI) in 2022 by PCR without any sequencing data.

The timeline between the last vaccination and blood collection across three distinct groups categorized by their booster vaccination status is depicted in Figure 1. The average duration from the last vaccination dose to the subsequent blood sample collection was 25.11 weeks, 19.24 weeks, and 16.9 weeks for the NB, single booster (SB), and DB groups, respectively. The statistical analysis using the Kruskal–Wallis test confirmed significant differences in the mean duration among these groups (*p* < 0.0001). Further examination using Dunn’s multiple comparison tests highlighted significant disparities when comparing the SB and DB groups to the NB group (*p* = 0.003 and <0.0001, respectively). Notably, while booster vaccinations contributed to variations in the timeline, the mean duration between the SB and DB groups was not significantly different (*p* = 0.63).

Among the participants, only 22 (19.46%) were confirmed to have contracted COVID-19 beyond 2021 (Table 3 and Figure 1). The mean duration from the confirmation of infection status to blood sampling was 21.72 (±1.45) months. Specifically, 81% of the confirmed cases were health workers, comprising 50% community health center staff and 31.81% hospital staff. The remaining 18.18% worked in a health-service-unrelated environment. Interestingly, most confirmed infection groups (68.18%) had received a second booster dose. Other participants who had never undergone a confirmatory test were categorized as uninfected and were referred to as untested individuals.

### 2.2. Laboratory Analysis

Before storage, the samples underwent heat inactivation to remove non-specific inhibition effects. This involved incubation at 56 °C in a circulating water bath for 30 min, followed by centrifugation at 15,000 rpm for 5 min at 4 °C. After heat inactivation, the samples were stored at −80 °C for subsequent analyses. All samples were subject to indirect ELISA for antibody titer assessment and pseudo-virus neutralizing assay for neutralization capacity (NC) evaluation.

Indirect ELISAs were performed using commercially available, HPLC-verified Wild-Type (WT) and XBB.1.5 RBD proteins as antigens (Sino Biological, Kawasaki City, Japan; #40591-V08H for WT and #40592-V08H146 for XBB.1.5). The antibody titers are expressed as the optical density (OD) of the serum samples. As previously described, the same ELISA procedure was employed to measure the ODs, reflecting each sample’s antibody (Ab)-binding capability to the respective antigens [15,16]. The 96-well microplates (Corning, Sakai, Japan, #3590) were coated with 0.2 μg/mL of either SARS-CoV-2 WT or the XBB.1.5 antigen and incubated overnight at 4 °C. Before incubation with the sera samples, the plates were blocked with 1% bovine serum albumin (BSA) in PBS (pH 7.4) for 1 h and then washed with PBS-T. The sera were diluted 1:100 in PBS containing 1% BSA. After incubation with sera for an hour, the plates were washed and then incubated with horseradish peroxidase (HRP)-conjugated monoclonal antibodies recognizing an Fc domain of human IgG for another hour. After incubation, the plates were rewashed, 100 μL/well of the substrate was added to each well, and the plates were incubated for 30 min at room temperature for color development. The absorbance was measured at 414 nm using a microplate reader, with results expressed as the Ab titers.

In the neutralization assay, the study emphasized assessing the overall trends in serological status rather than determining the exact neutralizing antibody (NAb) titers of individual samples. A brief assay can effectively correlate Ab OD with viral neutralization [15]. We utilized a VSV-based pseudo-virus expressing the Wuhan-Hu spike protein from earlier research [15,16]. Additionally, we engineered a pseudo-virus for this study that expresses the Omicron XBB.1.5 spike protein on its surface. To facilitate detection, we incorporated the luciferase gene into the viral genome. In the 96-well plate, 2 µL of serum was diluted in 48 µL of medium, followed by the addition of 50 µL of medium containing the pseudo-virus (3.2 × 10^4^ TCID_50_/well). Three wells without serum served as the controls. After incubation at 37 °C for 1 h, 100 µL of medium with 293T/hACE2+hTMPRSS2 cells (2 × 10^4^ cells/well) was added. Following a 24-h incubation period, the ONE-Glo EX™ Luciferase Assay System (Promega, Madison, WI, USA) was used to determine the percentage of viral internalization by comparing the luminescence of the serum-treated wells to that of the control wells. The serum’s NC was expressed as 100% minus the viral internalization percentage, reflecting the serum’s ability to neutralize the virus.

### 2.3. Data Analysis

Statistical analysis was performed using GraphPad Prism version 10.0 for Mac OS. Kruskal–Wallis test with a post hoc Dunn’s multiple comparison test was used for group analysis. A comparison between the two groups was made using the Mann–Whitney U test. Spearman correlation analysis and non-linear regression analysis were used to analyze the correlations between the Ab titers and the percentage of internalization. A *p*-value of <0.05 was considered statistically significant. 

## 3. Results

### 3.1. Evaluation of the Persistence of Antibodies against Wild-Type SARS-CoV-2 and XBB.1.5

We first measured serum Ab titers against the RBD of the original WT and Omicron XBB.1.5 using ELISA to investigate the effect of booster doses and infection on long-term humoral immunity against COVID-19 (Figure 2). As expected, individuals with booster doses demonstrated higher Ab titers against the WT in a stepwise pattern (Figure 2A). The DB group showed the highest Ab titers, whereas the NB group showed the lowest titers (NB vs. SB vs. DB = 0.48 ± 0.03 vs. 0.54 ± 0.05 vs. 0.73 ± 0.03, respectively). Interestingly, the anti-XBB.1.5 titer was higher than anti-WT titer in all subjects, yet the difference was not significant between the booster groups (Figure 2B). The mean anti-XBB.1.5 titer among the NB, SB, and DB groups was 1.43 ± 0.05, 1.51 ± 0.06, and 1.52 ± 0.06, respectively. The vaccines failed to yield a significant increase in Ab titers against both strains in individuals with BTI (Figure 2C,D). Individuals with BTI exhibited an average anti-WT titer of 0.67 ± 0.05, while those without BTI showed a titer of 0.59 ± 0.02. On the other hand, the anti-XBB.1.5 RBD titer in individuals with a confirmed BTI was 1.50 ± 0.05, and that in the untested individuals was 1.49 ± 0.04. Further analysis showed that the Ab titers against WT and XBB.1.5 among individuals with pre-infection were not significantly different from those without pre-infection (Figure 2E,F).

The serum Ab titers against XBB.1.5 are higher (1.507 ± 0.03) than those against WT in all subjects (0.606 ± 0.02) (*p* < 0.0001) (Appendix A). The subgroup analysis based on the number of vaccines received revealed consistently higher XBB.1.5 Ab titers relative to WT across all subgroups (Appendix A; NB = 1.438 ± 0.05 vs. 0.48 ± 0.03; SB = 1.513 ± 0.06 vs. 0.549 ± 0.05; DB = 1.520 ± 0.06 vs. 0.737 ± 0.03), with *p* < 0.0001 observed in each subgroup. Similarly, based on infection status, the level of anti-WT and anti-XBB.1.5 Ab titers is significantly different (Appendix A). Anti-XBB.1.5 Ab titers surpassed WT Ab titers in the pre-infected individuals (1.5 ± 0.05 vs. 0.67 ± 0.05, *p* < 0.0001) and in the untested group (1.49 ± 0.04 vs. 0.59 ± 0.02, *p* < 0.0001).

### 3.2. Natural Infection Results in a Better Neutralization Capacity (NC)

Next, we investigated the serum Ab NC against WT and XBB.1.5 VSV pseudo-virus. As expected, the NC against Omicron XBB.1.5 was lower than that against the WT. While the number of booster doses did not significantly impact NC against WT (Figure 3A), individuals with two booster doses exhibited notably higher NC against XBB.1.5 pseudo-virus compared with those receiving one dose of booster or none (Figure 3B). Across the three booster groups, NC against WT remained comparable, with nearly 100% neutralization observed (96.22 ± 1.67% for NB, 96.88 ± 1.64% for SB, and 97.93 ± 0.55% for DB).

An increased NC against XBB.1.5, along with the increased number of booster doses, was observed, although the increment was not statistically significant. Specifically, the NC against XBB.1.5 among the NB, SB, and DB groups were 80.71 ± 3.9%, 74.29 ± 6.7%, and 67.2 ± 6.3%, respectively. Given that NC for WT revealed nearly 100% neutralization for the majority of the sample, the NC was parallel between those with and those without BTI (Figure 3C).

A significant difference in NC against XBB.1.5 was demonstrated by individuals with BTI compared with those without (90.29 ± 3.89% vs. 71.75 ± 3.63%, respectively; Figure 3D; *p* < 0.05). The individuals with more booster doses demonstrated higher NC, especially those without BTI; however, a different case was shown with the pre-infected individuals, in whom the second booster did not result in antibodies with better NC (Figure 3E,F). The prevalence of pre-infected individuals with more than 50% neutralization of XBB.1.5 reduced from 100% to 80% after the second booster dose. On the other hand, the prevalence of uninfected individuals with more than 50% neutralization was 71%, 76%, and 84% in the NB, SB, and DB groups, respectively.

Appendix A clearly shows a comparative analysis of NC based on the strains of SARS-CoV-2. Evidently, NC against XBB.1.5 is lower than NC against WT in all subjects.

### 3.3. Correlation between Serum Antibody Titers and Neutralization Capacity for Wild-Type and Omicron XBB.1.5 Variants

We analyzed the relationship and model fit between Ab titers and their corresponding NC, as illustrated in Figure 4. Additionally, we examined the correlation and predictive score of Ab NC for the other variant. The Ab titers against the WT variant demonstrated a moderate negative correlation with their corresponding WT pseudo-virus internalization, as depicted in Figure 4A (r = −0.3155, *p* < 0.0001). A stronger cross-neutralization correlation was observed for the Omicron XBB.1.5 variant (r = −0.439, *p* < 0.0001), as shown in Figure 3C. A regression analysis revealed a minimal predictive relationship between WT Ab titers and their NC for both WT and XBB.1.5 variants (R^2^ = 0.064 and R^2^ = 0.125, respectively).

The antibodies targeting the Omicron XBB.1.5 variant (Figure 4B) also exhibited a moderate correlation with NC against their corresponding XBB.1.5 variant (r = 0.3499, *p* < 0.001). However, cross-neutralization capacity (cross-NC) against the WT variant showed no significant correlation, likely due to the low level of viral internalization demonstrated by all sera (Figure 3D) (r = −0.182, *p* = 0.053). The correlation and cross-correlation analysis displayed a limited predictive value (R^2^ = 0.1 and R^2^ = 0.069, respectively).

A subgroup analysis assessed the in-depth correlation between serum Ab titers and NC based on vaccination status and BTI history (Figure 3A–E). In the booster subgroup analysis, significant correlations were observed between Omicron XBB.1.5 Ab titers and NC in the NB group (Figure 3A, r = −0.357, *p* = 0.035) and SB group (Figure 3B, r = −0.553, *p* < 0.001). A regression analysis revealed moderate relationships between Ab titers and NC in these groups (R^2^ = 0.152 and R^2^ = 0.311, respectively). However, no significant correlation was found in the DB group (Figure 3C, r = −0.146, *p* = 0.32), with a meager regression score (R^2^ = 0.003).

Conversely, for the WT variant, low and non-significant correlations were observed in both the NB (Figure 3A, r = −0.106, *p* = 0.53) and SB groups (Figure 3B, r = −0.217, *p* = 0.24). However, a significant correlation was detected in the DB group (Figure 3C, r = −0.404, *p* < 0.01), accompanied by a higher regression score (R^2^ = 0.26) compared to the Omicron XBB.1.5 variant.

Moreover, individuals with BTI beyond 2021 exhibited similar NC for both the WT and Omicron XBB.1.5 variants (Figure 3D), resulting in comparable correlations between the serum Ab titers and NC in this group (r = −0.272, *p* = 0.25 and r = −0.168, *p* = 0.49, respectively). However, a notably better regression score was observed against Omicron XBB.1.5 (R^2^ = 0.476); moreover, the untested individuals displayed significantly moderate correlations between Ab titers and NC against Omicron XBB.1.5 (Figure 3E; r = −0.387, *p* < 0.0001). A regression analysis was only feasible for Omicron XBB.1.5 (R^2^ = 0.129), as the regression for WT was exceedingly low (R^2^ < 0.0001), which might be due to uniformly high NC against WT.

## 4. Discussion

Antibody titers against COVID-19, elicited from vaccination or infections, have been reported to decline over time and have become a growing concern [17,18,19]. Several studies suggest that vaccinations, particularly in individuals with pre-existing immunity, can initially boost Ab levels, gradually decreasing and stabilizing at lower levels over an extended period [4,20,21,22,23,24]. Our previous study on short-term immunity against the SARS-CoV-2 WT variant revealed that antibodies persisted at a high level at three months post-infection, yet no difference level between pre-infected individuals with uninfected vaccinated individuals was found [15]. In the current study, we evaluated the long-term immunity, at least one year ahead of the last vaccination, based on the number of booster doses and infection history beyond December 2021. Our current study confirms that booster doses have a pronounced effect on extending long-term immunity against WT and Omicron XBB.1.5 variants, particularly in individuals without infection history, as previously reported [25,26]. This is evidenced by the persistence of Ab titers and NC observed until 25 months after the last vaccination. Multiple reports suggest that BTI post-vaccination can significantly boost Ab titers, as evidenced by cases involving the Omicron variant infection following booster doses [20,27,28]. Recent studies indicate that individuals who received boosters and experienced BTI exhibit nearly double the amount of Ab titers compared to those without BTI, a phenomenon observed five months after receiving the booster [29]. However, in our study, we observed similar Ab titers between the post-infected individuals and the uninfected individuals. The Ab titer of the post-infected individuals, spanning approximately 21 months of the mean duration of the last known infections, may have spiked and decreased to eventually plateau at the same level as those of naïve vaccinated individuals when the blood sample was collected. The leveling off of the Ab titer may also be attributed to asymptomatic infections [30].

Neutralizing antibodies (NAbs) are pivotal in preventing SARS-CoV-2 infection, making the NAb activity assessment indispensable for addressing COVID-19 in diagnostic, therapeutic, and preventive contexts [31,32]. As NAb levels may decline over time and could be circumvented by viral mutations, continual monitoring of NAb activity is vital for guiding future prevention strategies [23,33]. Our study’s findings reveal that individuals who received booster vaccinations demonstrate enhanced long-term neutralization against the Omicron variant compared to those who did not, particularly those without BTI history. One plausible explanation for this trend is the efficacy of heterologous boosters, observed among our study participants, in bolstering neutralizing responses more effectively than homologous boosters. This approach to heterologous boosting may mitigate the off-target immunity induced by different vaccine types [22,34,35].

Antibodies against SARS-CoV-2 neutralize the virus in several mechanisms, including blocking the interaction of the RBD with ACE2 by binding to the RBD or binding with a co-receptor (TMPRSS2), thus blocking the subsequent steps [36]. Therefore, natural infection may induce more complex antibodies that may be not only capable of binding to the receptor, but also to the co-receptor. Vaccine-induced antibodies may not be as effective as naturally-induced neutralizing antibodies, as the latter may target the N-terminal domain of the spike protein [37]. Our data show that BTI increased NC against Omicron, which is particularly evident among those who received one booster dose, similar to previous reports [21,38]. Cohort studies investigating booster effects have reported a 4.1-fold increase in NAb response following the third dose, rising further to 7.1 times with booster doses in BTI cases compared to primary vaccine recipients [27]. BTI after booster administration is associated with sustained neutralizing antibodies, observed over 6 months to 2 years, and a decelerated rate of NAb waning [39,40,41]. However, in our observation, the individuals with BTI who received the second booster dose did not show a higher NC compared to those who received the first dose. Thus, we assume that two doses of inactivated virus vaccine boosted with one dose of mRNA vaccine is sufficient for a long-term NC among individuals with a history of infection.

Another interesting finding from our study is the near-complete neutralization against the WT strain of most samples, even of those obtained from individuals without a booster. Despite the expectation of a decrease in NC over time, several reports indicate the persistence of NC against WT, and even potential increases due to exposure to the mutant variants. It has been reported that NAb gained from a mutation may also help cross-neutralization to WT [42]. The correlation between Ab titers and their corresponding NC offers valuable insights into the immunity against COVID-19 and the potential severity of infections [4,43]. Cohort studies measuring Ab titers alongside neutralization activity have shown persistent correlations, with Ab titers maintaining a solid correlation with their neutralization response for at least five months until nine months post-infection [44,45]. However, the applicability of this correlation may be limited over time, as Ab titers eventually reach detectable plateaus while maintaining stable neutralization activity, as elucidated above. Despite a notable correlation between Ab titers and NC in our findings, with correlations hovering around 0.3 for both metrics, it is crucial to interpret these results cautiously in the context of long-term evaluations. Therefore, interpreting Ab titers and their associated neutralization activity necessitates careful consideration in long-term immunity assessments against COVID-19. In addition, a study reported that antibodies produced by B-cell clones against RBD do not always compete with ACE2 for RBD binding [46].

It has been suggested that neither infection nor vaccination alone could induce potent cross-neutralization against Omicron [47,48,49]. In the current study, all subjects received vaccination; therefore, we could not evaluate the impact of infection alone. A study reported that triple vaccination with an inactivated virus, even without BTI, demonstrated robust short-term cross-neutralization activity against both the Delta and Omicron variants [50]. In our study, the untested individuals with no confirmed BTI showed a lower neutralization activity than those with confirmed BTI. However, the NC against XBB.1.5 was still more than 70% in the naïve non-booster group, despite its waning due to longer duration than the other booster groups. This result is contradictive with a previous study showing almost complete XBB.1.5 evasion in three-dose-vaccinated sera after ten weeks [51] and another study showing that a two-dose vaccine produced antibodies with low avidity against the Omicron variant, but an additional dose increased the avidity [52].

This study has several limitations that warrant consideration. Firstly, its cross-sectional design, while providing rapid insights, may only partially capture the dynamics of long-term immunity. Thus, we could not evaluate the dynamic of the NC, the peak, or how long it was maintained at the same level. A prospective study would offer a more accurate assessment, not only for the dynamic of the immune response, but also in investigating infection history, which can sometimes go unnoticed during cross-sectional analyses. Thus, we prefer using the term “untested” for individuals without any confirmed infection. Moreover, we did not have data showing which SARS-CoV-2 strain circulated in Makassar, due to limited sequencing. Another limitation of this study was that the sample size was quite small. There was only one subject who had a confirmed infection in the no-booster group. Thus, we did not have sufficient data for a comparison between uninfected individuals and infected individuals in this group. Finally, we did not consider the variability of sex, age, body mass index, and smoking status of the study subjects, which were reported to correlate with the Ab-neutralizing activities [53]. Despite the limitations, this study has provided important information to guide future policies on COVID-19 vaccination in Indonesia. A two-dose whole-inactivated vaccine induced long-term immunity and sufficient cross-neutralization against mutant variants of SARS-CoV-2 for at least two years after vaccination, enhanced by booster doses in uninfected individuals.

## 5. Conclusions

The COVID-19 vaccine induced long-term immunity and cross-neutralization against the mutant variant of SARS-CoV-2 for at least 2 years post-administration in Indonesian residents. Individuals without booster doses maintain sufficient neutralizing capacity against the XBB.1.5. Booster doses enhanced the neutralization capacity, especially among naïve vaccinated individuals.

## Figures and Tables

**Figure 1 antibodies-13-00072-f001:**
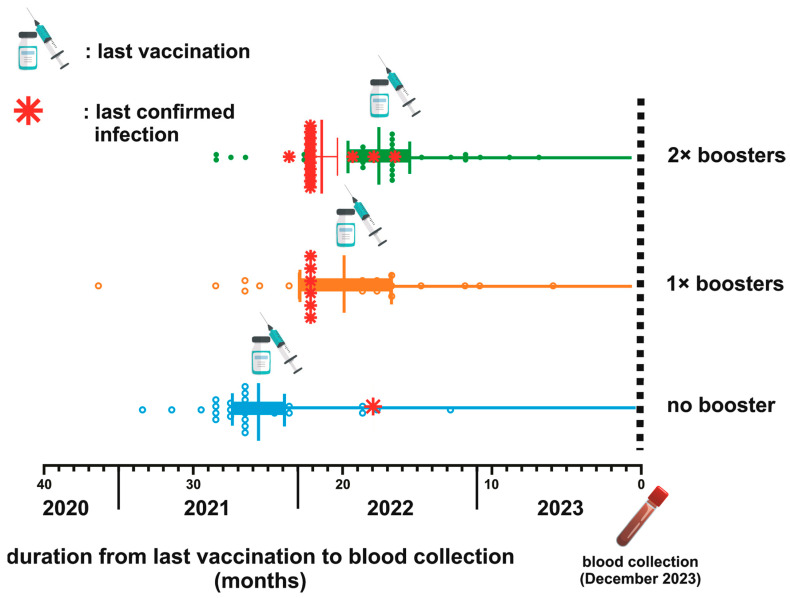
Schematic depiction of the vaccination timeline and breakthrough infections (BTI) for three groups based on booster status. The mean duration from the last vaccination to blood collection is 25.11 weeks for the no booster (NB) group, 19.24 weeks for the 1× booster (SB) group, and 16.9 weeks for the 2× booster (DB) group (*p* < 0.0001).

**Figure 2 antibodies-13-00072-f002:**
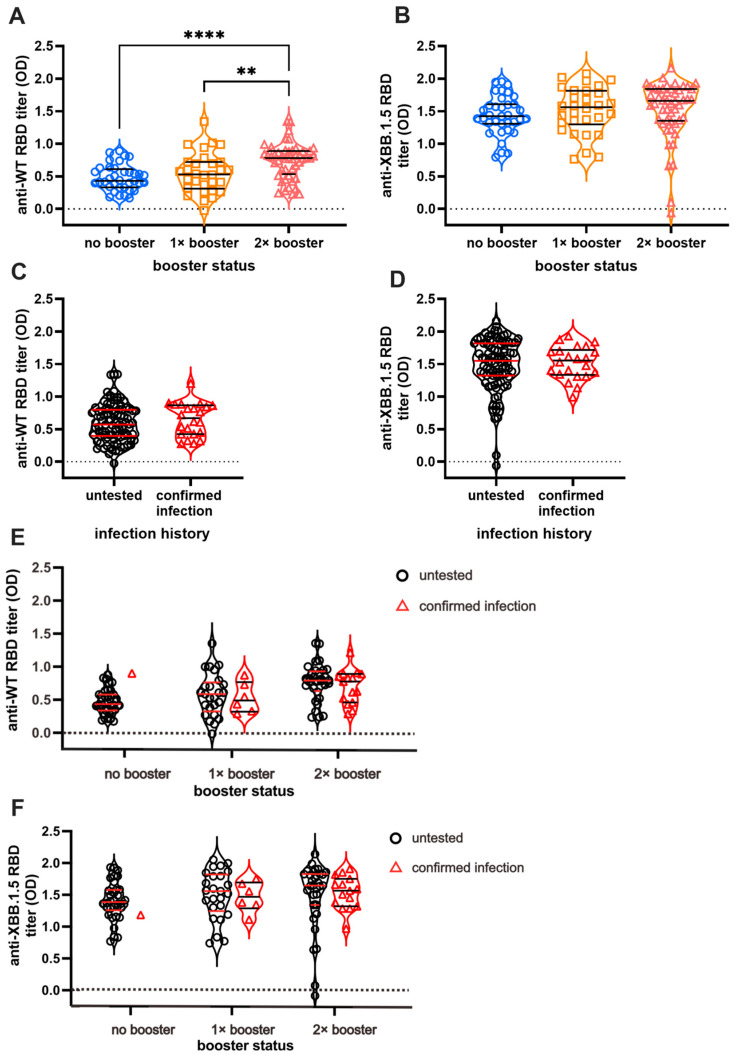
Anti-RBD IgG titer of SARS-CoV-2 Wild-Type (WT) and Omicron XBB.1.5. While the booster doses affect the persistence of antibody (Ab) titers against SARS-CoV-2 WT (**A**), they do not affect the persistence of anti-XBB.1.5 Ab titer (**B**). Breakthrough infections (BTI) resulted in higher titer against WT SARS-CoV-2 (**C**) but no significant differences in anti-XBB.1.5 titer compared to non-infected participants (**D**). The Ab titers based on infection status and vaccine are shown in (**E**) and (**F**). The Ab titers of individuals with BTI beyond 2021 against WT (**E**) and XBB.1.5 (**F**) are not significantly different from vaccinated individuals without BTI. The antibody titers were measured using indirect ELISA and are expressed as ELISA’s optical density (OD) measurements at 414 nm. Individual values are shown, and horizontal lines represent median and quartile of Ab titers. Statistical analyses were conducted using the Mann–Whitney test: **** *p* < 0.0001, ** *p* < 0.01.

**Figure 3 antibodies-13-00072-f003:**
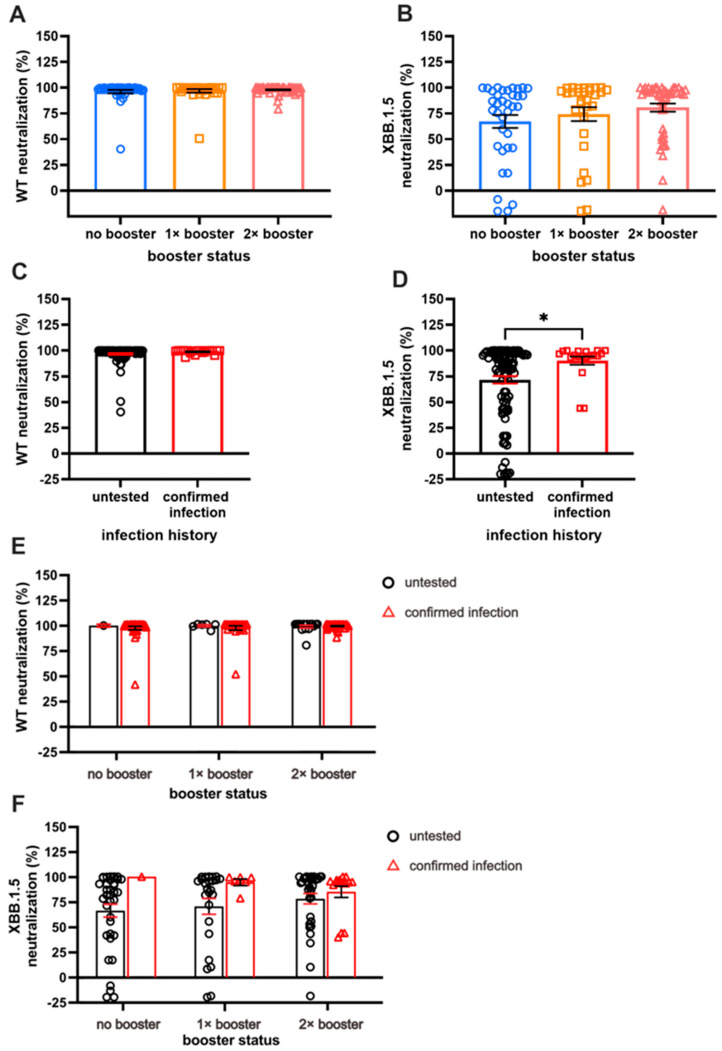
A comparison of long-term neutralization capacity (NC). (**A**) Almost-complete NC against WT in all subjects. (**B**) Comparison of NC against XBB.1.5, with double booster (DB) groups having the highest NC compared to no booster (NB) and single booster (SB) groups. (**C**) Individuals with breakthrough infection (BTI) exhibit comparable NC against WT compared to the untested group. (**D**) Individuals with BTI demonstrate higher NC against XBB.1.5 than untested groups. (**E**,**F**) The effect of booster doses on WT NC and XBB.1.5 NC among individuals with confirmed BTI and those without BTI. Serum NC was measured using ONE-Glo EX^TM^ Luciferase Assay System. Individual values are shown with bars that represent the means and horizontal lines that represent SEM. Statistical analyses were performed using the Mann–Whitney U test, * *p* < 0.05.

**Figure 4 antibodies-13-00072-f004:**
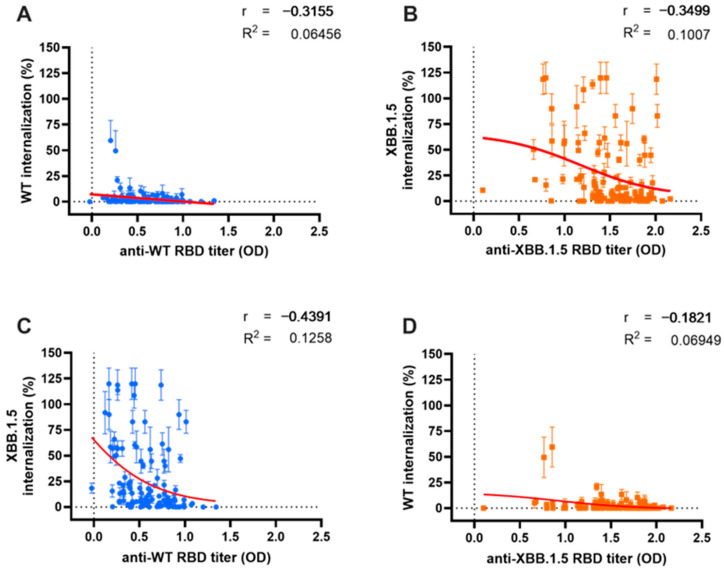
Correlation between antibody (Ab) titer and either neutralization capacity (NC) or cross-neutralization. A similar weak relationship between serum Ab titers and their respective NC was observed for both Wild-Type (WT) (blue dots) (**A**) and Omicron XBB.1.5 (orange dots) (**B**). Cross-correlation analysis between WT Ab titers and Omicron XBB.1.5 internalization (**C**); and Omicron XBB.1.5 Ab titers and WT internalization (**D**). Serum Ab titers were measured using indirect ELISA and are expressed as optical density at 414 nm, while serum NC was measured with the ONE-Glo EX™ Luciferase Assay System. Red lines depict the non-linear regression model between variables.

**Table 1 antibodies-13-00072-t001:** Grouping of study subjects based on their booster status.

Group	Vaccine Type		
	Primary Vaccine	1st Booster	2nd Booster
No booster	CoronaVac	-	-
1-time booster	CoronaVac	BNT162b2	-
2-times booster	CoronaVac	BNT162b2	mRNA1273

**Table 2 antibodies-13-00072-t002:** Distribution of study subjects by vaccination status.

Group *	*n* (%)	Sex		Age; Median (Interval; y.o)	Infection beyond 2021	
		Male; *n* (%)	Female; *n* (%)		Confirmed (%)	Untested ^#^ (%)
No booster (NB)	37 (32.74)	13 (35.13)	24 (64.86)	19.66 (17–23)	2/37 (5.4)	35/37 (94.59)
1-time booster (SB)	29 (25.66)	7 (24.13)	22 (75.86)	22.00 (17–53)	5/29 (17.24)	24/29 (82.75)
2-times booster (DB)	47 (41.59)	11 (23.4)	36 (76.59)	34.96 (20–40)	14/47 (29.78)	33/47 (70.21)

* NB: No booster; SB: Single booster; DB: Double booster. ^#^ subjects without any COVID-19 confirmatory test.

**Table 3 antibodies-13-00072-t003:** Characteristics of study subjects based on infection status.

Infection Status	*n* (%)	Sex		Age; Median (Interval; y.o)	Employment		Vaccination Status *			Duration from Last Infection in 2022; Mean ± SD (Months)
		Male; *n* (%)	Female; *n* (%)		Health Worker *n* (%)	Others; *n* (%)	NB; *n* (%)	SB; *n* (%)	DB; *n* (%)	
Confirmed infection beyond 2021	22 (19.46)	1 (4.54)	21 (95.45)	33.23 (17–59)	18 (81.81)	4 (18.18)	1 (4.54)	6 (27.27)	15 (68.18)	21.72 (1.45)
Untested ^#^	91 (80.53)	30 (32.96)	61 (67.03)	21.83 (17–59)	29 (31.86)	62 (68.13)	35 (38.46)	23 (25.27)	32 (35.16)	-

* NB: No booster; SB: Single booster; DB: Double booster. ^#^ subjects without any COVID-19 confirmatory test.

## Data Availability

The raw data supporting the conclusions of this article will be made available by the authors upon request.

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
