# Peer review of "Long-Term Immunity against SARS-CoV-2 Wild-Type and Omicron XBB.1.5 in Indonesian Residents after Vaccination and Infection"

_2073-4468, 2024, doi:10.3390/antib13030072_

Round 1

Reviewer 1 Report

Comments and Suggestions for Authors

The authors evaluated long-term immunity against WT and XBB.1.5 in 113 CoronaVac-immunized people with different vaccination status in Indonesia. They compared anti-RBD IgG and NAb titers against WT and XBB.1.5. This is a unique study design and showed interesting data against recent SARS-CoV-2 variant. Specific comments follow.

Major points:

1.     Line 112: Neither reference 15 nor 16 describing NAb assay method. Please describe the method in detail such as serum heat-inactivation treatment to avoid non-specific inhibition and serum dilution method. (Lines 116-117: What do you mean by “final dilution rate”?

2.     Line 121: NAb titer should be expressed in IC50 dilution. Then you will see the difference among groups in Figure 3 and should see the good correlations in Figure 4.

3.     Please justify the meaning to compare “untested” and “confirmed infection” groups. I think proper comparison should be anti-NP antibody Negative and Positive groups.

Minor points:

1.     Please define abbreviations only when it appears first. For example, “wild-type (WT)” appears more than ten times in the manuscript.

2.     Line 337: “Nab” should be “NAb”.

Author Response

  1. Line 112: Neither reference 15 nor 16 describing NAb assay method. Please describe the method in detail such as serum heat-inactivation treatment to avoid non-specific inhibition and serum dilution method. (Lines 116-117: What do you mean by “final dilution rate”?

    Response 1

    Thank you for pointing this out. We agree with your comment and have revised the manuscript accordingly. In our previous study (Ref [16]), we initially calculated the IC50 values. We then modified and optimized this calculation method in our subsequent study, as documented in Ref [15].
    We have added a detailed description of the heat-inactivation method at lines 137-140 to clarify this procedure as follows:

    ” Before storage, samples underwent heat-inactivation to remove non-specific inhibition effects. This involved incubation at 56°C in a circulating water bath for 30 minutes, followed by centrifugation at 15,000 rpm for 5 minutes at 4°C. After heat inactivation, samples were stored at -80°C for subsequent analyses. ”

    The term "final dilution rate" refers to the final serum concentration within the mixture of serum, virus, and cells. Specifically, the "serum final dilution rate" denotes the optimized concentration of serum in each well, as established in previous studies. This concentration was chosen to effectively illustrate overall trends in serological status during the NAb assay. However, to avoid confusion, we rewrote the NAb assay method. These changes can be found in lines 164-172:

    “…..In the 96-well plate, 2 µL of serum was diluted in 48 µL of medium, followed by the addition of 50 µL of medium containing the pseudovirus (3.2 x 104 TCID50/well). Three wells without serum served as controls. After incubation at 37°C for 1 hour, 100 µL of medium with 293T/hACE2+hTMPRSS2 cells (2 x 104 cells/well) was added. Following a 24-hour incubation period, the ONE-Glo EX™ Luciferase Assay System (Promega, Madison, WI, USA) was used to determine the percentage of viral internalization by comparing the OD from the luciferase assay to the control wells. The serum’s neutralization capacity (NC) was expressed as 100% minus the viral internalization percentage, reflecting the serum’s ability to neutralize the virus.”

    1. Line 121: NAb titer should be expressed in IC50 dilution. Then you will see the difference among groups in Figure 3 and should see the good correlations in Figure 4.3.     Please justify the meaning to compare “untested” and “confirmed infection” groups. I think proper comparison should be anti-NP antibody Negative and Positive groups.

    Response

    Thank you for your insightful comment. We have made revisions to the Methods section, particularly in the neutralization assay method, to emphasize this point. Our study prioritizes the analysis of overall trends in serological status, focusing less on precise neutralizing antibody (NAb) titers for individual samples. This methodology is based on previous studies that have effectively demonstrated the relationship between antibody titers and viral neutralization (see ref. [15]). We have also included additional details and modifications regarding our neutralization assay method in lines 157-163 to enhance the clarity and precision of our methods, as follows:

    “In the neutralization assay, the study emphasized assessing overall trends in serological status rather than determining the exact neutralizing antibody (NAb) titers of individual samples. Previous studies have demonstrated that a brief assay can effectively correlate Ab OD with viral neutralization[15]. We utilized a VSV-based pseudovirus expressing the Wuhan-Hu spike protein from earlier research[15,16]. Additionally, we engineered a pseudovirus for this study that expresses the Omicron XBB 1.5 spike protein on its surface. To facilitate detection, we incorporate the luciferase gene into the viral genome “

    In this study, participants were categorized as 'untested' if they had never undergone any confirmatory COVID-19 test. We now emphasize this point in the methods section as follows:

    “Other participants who had never undergone a confirmatory test were categorized as uninfected and were referred to as untested individuals.” (lines  129-130)

     Given that asymptomatic and mild infections were less frequently documented after 2022, the 'untested' group may include individuals who had undiagnosed infections. Thus, we considered this as one of the limitations of the study, as we stated in the discussion as follows (lines 381-387):

    “This study has several limitations that warrant consideration. Firstly, its cross-sectional design, while providing rapid insights, may only partially capture the dynamics of long-term immunity. Thus, we could not evaluate the dynamic of the NC, the peak, and how long it is maintained at the same level. A prospective study would offer a more accurate assessment, not only for the dynamic of immune response but also in investigating infection history, which can sometimes go unnoticed during cross-sectional analyses. Thus, we prefer using the term “untested” for individuals without any confirmed infection…”

    While this grouping may introduce some variability, it still allows for meaningful comparisons of immune responses between those with and without confirmed infections. This approach provides valuable insights into immunity against COVID-19 reflecting the real condition of the community. We did not use anti-NP negative and positive since we did examine anti-NP antibody titer. However, our previous work (unpublished) has shown that even in naïve vaccinated individuals, anti-NP was high and comparable with those who have contracted natural infection of COVID-19.

     Minor points:

    1. Please define abbreviations only when it appears first. For example, “wild-type (WT)” appears more than ten times in the manuscript.

    Response

    Thank you for pointing this out. We agree with this comment. Therefore, we have accordingly revised the manuscript to emphasize this point. We mentioned the defined abbreviations only in the beginning of each section. The following are the detailed revisions:

    Abstract and figure’s captions: all abbreviations in the abstract sections figures caption have been evaluated and no change has been made.

    Section headings: We remove all abbreviations in section headings

    Body

    The following abbreviations are defined at their first occurrence in the manuscript:

    "Ab" for antibody, first defined at line 145.

    "WT" for Wild-Type, first defined at line 142.

    "DB" for double booster, first defined at line 94.

    "SB" for single booster, first defined at line 109.

    "NB" for no booster, first defined at line 95.

    "BTI" for breakthrough infection, first defined at line 100.

    "NC" for neutralization capacity, first defined at line 140.

    We have reviewed the entire manuscript and removed any repeated definitions of these abbreviations to avoid redundancy.

    1. Line 337: “Nab” should be “NAb”.

    Thank you for pointing this out. We agree with your comment and have revised the manuscript accordingly. We have corrected the abbreviation from “Nab” to “NAb” to maintain consistency throughout the text.

Reviewer 2 Report

Comments and Suggestions for Authors

The manuscript presents an assessment of long-term humoral immunity to wild-type SARS-CoV-2 and Omicron XBB 1.5 in Indonesian residents after vaccination and infection.   Comments:   Line 17—Was the aim of the study to assess immunogenicity (strength of activation) or to assess the durability of the humoral response?   Line 98-100 —Sentence not very understandable. You might think that the HEK293 protein was used. It is worth checking "obtained in human embryonic kidney"   Line 138-139 — how was the infection confirmed? please provide the name of the tests.   Table 2 - what was the gender distribution in the study groups?   Line 130-167 - data without results should be moved to "2.1. Study design and study subject. The materials and methods should include the characteristics of the study population.   Line 169-184 - please provide information in the text about the units in which titeres is expressed, e.g. in line 174.   Figure 2, E, F - please mark more clearly: untested, infected.   It is better to replace "booster shots" in the entire article with booster doses.

Comments on the Quality of English Language

Reconsider after major revision

Author Response

comment 1: Line 17—Was the aim of the study to assess immunogenicity (strength of activation) or to assess the durability of the humoral response?   

Reply:
Thank you for pointing this out. Our aim was to assess the durability of the humoral response, as we evaluated the immune response over time rather than the immediate strength of activation following vaccination or infection. We have revised lines 16-17 accordingly to reflect this focus, as follows:

” This study aimed to assess long-term humoral immune response in sera collected in Makassar …….”

comment 2: Line 98-100 —Sentence not very understandable. You might think that the HEK293 protein was used. It is worth checking "obtained in human embryonic kidney"  

Reply:
Thank you for pointing this out. We agree with your comment and have revised lines 141-143 (the current version) accordingly to reflect this focus, as follows:

” Indirect ELISAs were performed using commercially available, HPLC-verified Wild-Type (WT) and XBB 1.5 RBD proteins as antigens (Sino Biological; #40591-V08H for WT and #40592-V08H146 for XBB 1.5). ”

 comment 3: Line 138-139 — how was the infection confirmed? please provide the name of the tests.   Table 2 - what was the gender distribution in the study groups?   

Reply:
Thank you for pointing this out. We agree with your comment and have revised the manuscript to emphasize this by adding the COVID-19 test method at lines 99-100, as follows:

” We recorded the confirmed breakthrough infection (BTI) in 2022 by PCR…. ”

Regarding the gender distribution, as shown in the table 2, there were more female participants than males in each group.

comment 4: Line 130-167 - data without results should be moved to "2.1. Study design and study subject. The materials and methods should include the characteristics of the study population.   

Reply:
Thank you for pointing this out. We agree with this comment. Therefore, we have accordingly revised the manuscript to emphasize this point by moving the data without results at previously lines 130-167 to line 92-130

comment 5: Line 169-184 - please provide information in the text about the units in which titers is expressed, e.g. in line 174.   

Reply:
Thank you for pointing this out. We agree with your comment and have revised the manuscript to clarify the antibody measurement method using serum antibody optical density. Lines 143-144 were revised to clarify the measurement method as follows

“The antibody titers are expressed as the optical density (OD) of serum samples.”

  • Figure 2 Caption has been revised to reflect the clarification in the measurement method.

Figure 2. Anti-RBD IgG titer of SARS-CoV-2 Wild-Type (WT) and Omicron XBB 1.5. While the booster doses affect the persistence of antibody (Ab) titers against SARS-CoV-2 WT (A), they do not affect the persistence of anti-XBB 1.5 Ab titer (B). Breakthrough infections (BTI) resulted in higher titer against WT SARS-CoV-2 (C) but no significant differences in anti-XBB 1.5 titer (D) compared to non-infected participants. The Ab titers based on infection status and vaccine are shown in E and F. The Ab titer of individuals with BTI beyond 2021 against WT (E) and XBB 1.5 (F) are not significantly different from vaccinated individuals without BTI. The antibody titers were measured by indirect ELISA and are expressed as ELISA's optical density (OD) measurements at 414 nm. Statistical analyses were conducted using the Mann-Whitney test: **** p<0.0001, ** p<0.01.

  • Figure 4 Caption has been revised accordingly as follows

Figure 4. Correlation between antibody (Ab) titer and either neutralization capacity (NC) or cross-neutralization. A similar weak relationship between serum Ab titer and their respective NC was observed for both Wild-Type (WT) (blue dots) (A) and Omicron XBB 1.5 (orange dots) (B). Cross-correlation analysis between WT Ab titers and Omicron XBB 1.5 Internalization (C) and Omicron XBB 1.5 Ab titers and WT Internalization (D). Serum Ab titer were measured by indirect ELISA and were expressed as optical density at 414 nm, while serum NC was measured by the ONE-Glo EX™ Luciferase Assay System. Red lines depict the non-linear regression model between variables.

comment 6: Figure 2, E, F - please mark more clearly: untested, infected.   It is better to replace "booster shots" in the entire article with booster doses.

Reply:
Thank you for pointing this out. We agree with your comment and have revised the manuscript accordingly. Specifically, we have changed the pattern in Figures 2.E, 2.F, 3E and 3F to use markers for better visualization. Additionally, we have replaced all instances of “shots” with “doses” for consistency.

Round 2

Reviewer 1 Report

Comments and Suggestions for Authors

The authors had adequately addressed all comments. One last comment:

Line 169: Luciferase Assay must be measured by luminescence but not by OD.

Author Response

Comments: The authors adequately addressed all comments. One last comment:Line 169: Luciferase Assay must be measured by luminescence but not by OD.

Response: Thank you for correcting this mistake. We have changed "OD"  with "luminescence" as corrected.

We are attaching the second revised manuscript. All changes as suggested by all reviewers are highlighted in green.

Reviewer 2 Report

Comments and Suggestions for Authors

The authors have taken into account all the comments, which has made the manuscript more understandable.

However, they have overdone the "OD values" throughout the article. It would be enough if the type of units was given in the materials and methods and graphs.

Author Response

Comment : The authors have taken into account all the comments, which has made the manuscript more understandable. However, they have overdone the "OD values" throughout the article. It would be enough if the type of units was given in the materials and methods and graphs.

response: Thank you for pointing this out. We have changed the "OD values" to "Ab titer", and we explained in the method section that the Ab titers are expressed as OD obtained in ELISA reading (line 114). We have also changed the figure legends and the graphs' labels accordingly. All changes as suggested by all reviewers are highlighted in green in the attached second revised manuscript.
